# Water Security and River Basin Revitalization of the São Francisco River Basin: A Symbiotic Relationship

**Larissa Alves da Silva Rosa** [1,*] , **Manuela Morais** [2] **and Carlos Hiroo Saito** [3]

1   Center for Sustainable Development, University of Brasília, Asa Norte, 70904-970 Brasilia, Brazil
2   ICT, Institute of Earth Sciences, University of Évora, Rua Romão Ramalho 59, 7000-671 Évora, Portugal; mmorais@uevora.pt
3   Department of Ecology, Institute of Biological Sciences, and Center for Sustainable Development, University of Brasília, Asa Norte, 70904-970 Brasilia, Brazil; carlos.h.saito@hotmail.com
*   Correspondence: larissa.rosas15@gmail.com

**Abstract:** What is river basin revitalization's place in relation to water security? This question is the basis of our reflection, posed to help in the understanding of the evolution of both concepts, taking management of the São Francisco River Basin (Brazil) as a case study. With this main objective in mind, a literature review was carried out, followed by the collection of survey data on the watershed's revitalization program. In this context, the members of the São Francisco River Basin Committee (a total of 124 participants) were consulted, using questionnaires with the Delphi method. The respondents (a total of 47) chose the river basin revitalization strategy as the main measure to achieving water security in the São Francisco River Basin. They also highlighted the importance of the environmental dimension, underlining measures for conservation and restoration of the ecosystem's natural functions. The concept map tool was adopted for a comparative perspective between conceptual implications of revitalization and water security for the studied river basin's conservation. The results showed the existence of a symbiotic relationship between both concepts. Consequently, we conclude that it is urgent to reconcile water use and ecosystem ecological integrity through the comprehensive concept of water security.

**Keywords:** river basin committee; Delphi method; water management; river restoration; semi-arid; Brazil

## 1. Introduction

"Water security for peace and development" is the theme of the 9th World Water Forum, scheduled to be held in 2021, in Dakar, the capital of Senegal, Africa. Among other objectives, the next Forum will be held to generate commitments and actions to provide water security around the world, aligned with Sustainable Development Goal 6. Therefore, countries should take advantage of this occasion to strengthen water security in Africa, the host continent, which is hit by frequent droughts and where one-third of the population do not have access to potable water [1]. With a similar concern, the Global Water Partnership (GWP) reports that a world with water security demands an articulation between the intrinsic value of water, its use, and human survival and well-being [2]. In fact, its recent 2020–2025 strategy places water security at the center of the debate, facilitating multistakeholder involvement for integrated and responsible water resource management [3]. This promotes the water security concept as a research topic of fundamental relevance.

At the 2nd World Water Forum (2000), the concept of water security was mainly focused on quantity and quality aspects. In this sense, Witter and Whiteford [4] defined water security as a condition of guaranteed sufficient water volume at a fair price and of adequate quality to meet human needs, concerning health and well-being at the local, regional, and national levels. This preliminary conceptualization does not consider the

environmental component, i.e., the conservation of natural ecosystems from the perspective of their own integrity and ecological functions.

According to the Ministerial Declaration of the 2nd World Water Forum (2000), water security means ensuring that: (i) freshwater ecosystems are protected and improved; (ii) sustainable development and political stability are promoted; (iii) each person has access to enough drinking water at an affordable cost to lead a healthy and productive life; and iv) vulnerable populations are protected against water-related risks [5]. This Forum already incorporates the environmental concern in its water security definition and respective Ministerial Declaration, albeit with less importance attached. Since then, the conceptual discussion has advanced and is now being debated in several countries, including Brazil. In fact, according to the available literature [6–15], the concept of water security is very centered on the human being (anthropogenic perspective), not compromising its respective uses (i.e., production of water for human consumption, industry, and agriculture), and maintaining water quality according to defined standards. In this context, monitoring programs have been defined with specific objectives, such as: definition of risk zones and definition of measures to prevent water pollution and contamination [16] to guarantee access to water in a human rights framework [14]. Still related to safety aspects, the construction of large infrastructures for water storage, sometimes with an international dimension, requires specific projects related to engineering construction and the materials utilized, addressing the potentially destructive impact that water can have. These potential impacts include degradation caused by extreme natural events, but also possible terrorist threats, which should not be overlooked [17]. Nevertheless, according to Lautze and Manthrithilake [10], water security is a broad concept with five components: (1) access to potable water for basic human needs or domestic use; (2) provision of water for productive activities; (3) environmental conservation or protection; (4) prevention of water-related disasters; and (5) provision of water for national security or independence. Consequently, unlike other natural resources, a lack of water is not the only threat to societies; it is also important to prevent the destructive potential of water, such as in uncontrolled flood conditions. From this perspective, water can pose a disaster in its absence (scarcity) or excess (floods).

In Europe, water security is intrinsically integrated into the Water Framework Directive 2000, which assesses the status (ecological and chemical) of water bodies [18]. In the Middle East, the discussion is focused on cross-border issues [19]. Bakker and Morinville [11] identified that commitment to good governance was a prerequisite for water security in Canada. In India, Narain et al. [20] discussed the social differences that shape unequal access to water. In Africa, Soyapi [21] correlated it with the human right to water. In Brazil, the discussion is mainly focused on major water infrastructure, as mentioned in the recently approved Water Security National Plan [22]. The Plan states that an ideal water security scenario should be planned, dimensioned, implanted, and managed properly, taking into account both the balance between supply and demand and contingency situations, as a result of vulnerability to extreme climatic events. However, the list of priorities for water security planning does not includes conservation and restoration measures, not even for aquatic zones, and is often referred to as river basin revitalization. This concept, containing a set of permanent and integrated actions for preservation, conservation, and environmental recovery, is aimed at the sustainable use of natural resources, the improvement of socioenvironmental conditions, and the availability of water in quantity and quality for multiple uses [23]. Therefore, theoretically, river basin revitalization is a premise for water security in any location. In practice, one can ask: What is river basin revitalization's place in relation to water security? How are they connected?

As a case study, we focused on the São Francisco River Basin (SFRB) due to its strategic importance for the country concerning territorial, political, socioeconomic, cultural, and environmental aspects. This river is the longest in the country, running entirely through the Brazilian territory, and 58 percent of its area is located in the Northeastern Drought Polygon of Brazil [24]. This semi-arid region is a geographical center that radiates government

initiatives, including projects to revitalize and transfer its waters to other basins, indicating that water security is a hot issue discussed in the SFRB.

This manuscript aims to analyze the interdependencies between the "water security" and "river basin revitalization" concepts, in the context of SFRB management. The knowledge of involved stakeholders (São Francisco River Basin Committee (SFRBC) members) was considered to address these conceptual interdependencies. The Delphi method was used, which is a qualitative data collection technique. Questionnaires were sent to SFRBC members in two consecutive rounds to obtain a consensus. Through the use of this participatory method, the study intended: (i) to investigate the perception and understanding of the SFRBC about "water security in the basin" and (ii) to analyze its relationship with "river basin revitalization", based on the hypothesis that river basin revitalization could be the main strategy to safeguard water security in other basins.

## 2. Materials and Methods

The research design consisted of three sequential and complementary steps, as shown in Figure 1: (i) a review of the literature to build a conceptual basis on water security and river basin revitalization; (ii) application of questionnaires to members of the SFRBC through the Delphi method to test the research hypothesis; and (iii) the production of synthetic conceptual modeling to identify the relationships between the concepts and the obtained knowledge from the stakeholders.

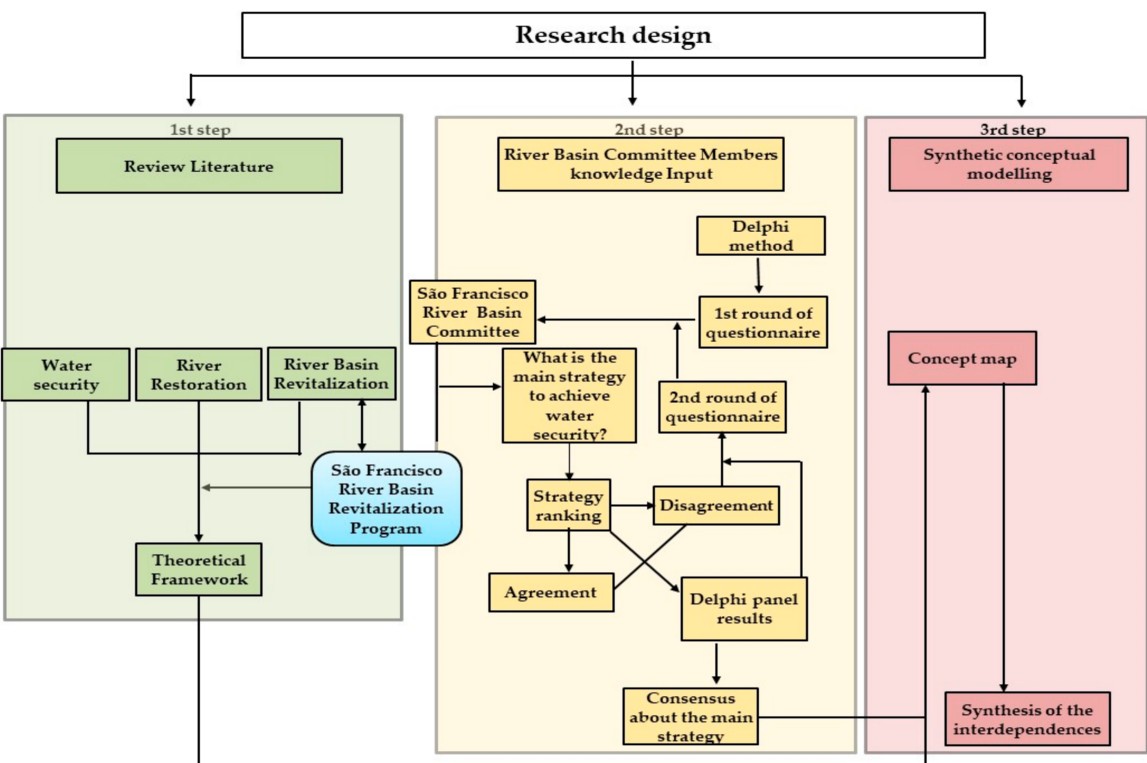

**Figure 1.** Research method design synthesis flowchart.

Within the first step, an exhaustive gathering of available information produced by societies, academies, and institutions was made, with regard to "water security", "river restoration", "river basin revitalization", and "the São Francisco River Basin Revitalization Program" (SFRP), without exhausting the possibilities of investigating the concepts applied to other river basins. The review of the literature built the theoretical framework that supported the research.

The second step consisted of connecting the theoretical framework with empirical data through a participatory approach using the Delphi method. The research cohort

chosen consisted of SFRBC members, who were consulted in two successive rounds of questionnaires, with the objective of identifying how they perceived the main strategies to achieve water security. Question Pro® software was used to send questionnaires online and to tabulate data.

Using the Delphi method, a panel of experts was consulted in order to obtain systematized answers that expressed their opinions on the topics of interest [25]—in the present SFRB, the relationship between water security and water management practices. The experts chosen to compose the panel had to be personally involved with the addressed question; have information and experiences relevant to the process; be motivated to participate and trust that the results would provide valuable information [26]. In the present research, the 62 full and alternate members that made up the SFRBC were selected, giving a total of 124 respondents. Of this total, 47 responses were obtained; that is, approximately 38% of the members answered the questionnaire in the first round. The educational level of the respondents had to be heterogeneous. Thus, the largest proportion of respondents had a master's degree, with 12 specialists (27% of the sample), followed by 10 specialists (23%) with higher education. Particularly in the case of the SFRBC, qualifications and technical knowledge were essential, improving social participation in the theme of water resource management in the basin. However, to participate in the Delphi method, a high degree of academic education is not necessary; it was enough that the respondents to the questionnaire were well informed about the area in question [27].

In the first round of the Delphi method, SFRBC members were asked to rank from 1 to 10, in order of priority, the 10 strategies previously defined that, in their opinion, could contribute to the improvement of water security, based on the available literature and technical reports, especially the National Water Security Plan [22] and the São Francisco River Basin Water Resources Plan [28]. For each of the 10 strategies, a frequency distribution was determined: how many times a strategy was chosen first, second, third, or fourth. From this, its importance was inferred. The data were summarized, indicating how many times each strategy was mentioned as the most important and the average value of the mentions.

Sequentially, these statistical data were forwarded to the experts again, and then they were asked to, if deemed pertinent, review their positions before the group's opinion was finalized. For the second round of questionnaires, the answers were transformed into questions, and the respondents were asked to review their agreement with the results of the first round. The second-round questionnaire was sent only to the 47 members who validly responded to the first round. Of these 47 initial participants, only 26 participated in the second round, representing a rate of return of 57%. Within the sequential process, the experts had the opportunity to find out the opinions of their peers, being able to review their position and justify their responses over two rounds of consultations, which favored convergence and reaching consensus on the issues addressed. Although consensus is an objective of the Delphi method and, consequentially, of this research, the variability of responses and points of view about the object in analysis is also important. To verify consensus in the responses, some authors suggest a high rate of representativeness, which corresponds to 80% of the responses of members concentrated in one category [29]. This value was adopted here.

In the third step, the concept maps tool was utilized in order to obtain a synthesis of the interdependences between the concepts of water security and river basin revitalization [30]. Concept maps are useful as a template or support to organize concepts and information in a schematic way, visualizing the chain of processes from an interdisciplinary perspective, being able to show the interfaces, complementarities, and antagonisms between different theoretical references of the same problem [31]. In this step, our intention was to include relationships identified by SFRBC members and analyze the diagram as a systemic integrated whole view, placing the concept of river basin revitalization within the water security framework and highlighting the relationships between them. The Cmap Tools® software was used to build this concept map.

## 3. Results and Discussion

### 3.1. The View of the Basin Committee on the Role of Revitalization

Table 1 presents the ranking of the average grades, attributed by SFRBC members on the strategies that contribute to the improvement of water security of the SFRB.

**Table 1.** Frequency distribution related to the strategy's hierarchy obtained in the first phase of application of the questionnaires to members of the São Francisco River Basin Committee (SFRBC) (*n* = 47), with values >50 in bold.

| | Strategies | Obtained Grades Frequency (%) | | | | | | | | | | Weighted Average |
|---|---|---|---|---|---|---|---|---|---|---|---|---|
| | | 1 | 2 | 3 | 4 | 5 | 6 | 7 | 8 | 9 | 10 | |
| **1** | River Basin Revitalization | 0.22 | 0.00 | 0.11 | **0.53** | 0.22 | 0.06 | 0.00 | 0.02 | 0.04 | 0.00 | 3.70 |
| **2** | Implement Water Pact | 0.04 | **0.62** | 0.04 | 0.00 | 0.02 | 0.06 | 0.08 | 0.02 | 0.04 | 0.06 | 3.72 |
| **3** | Increase the efficiency of production processes | 0.44 | 0.02 | 0.11 | 0.08 | 0.04 | 0.04 | 0.02 | 0.02 | 0.08 | 0.11 | 3.91 |
| **4** | Increase resilience in the face of hydrological disasters (droughts or floods) | 0.02 | 0.06 | **0.53** | 0.04 | 0.11 | 0.00 | 0.02 | 0.06 | 0.04 | 0.11 | 4.59 |
| **5** | Invest significantly in sanitation improvement | 0.11 | 0.00 | 0.08 | 0.08 | **0.51** | 0.08 | 0.04 | 0.04 | 0.02 | 0.04 | 4.91 |
| **6** | Construction of Water Infrastructure | 0.06 | 0.06 | 0.04 | 0.08 | 0.08 | 0.48 | 0.04 | 0.06 | 0.06 | 0.02 | 5.49 |
| **7** | Intensify Environmental Inspections | 0.00 | 0.06 | 0.04 | 0.06 | 0.06 | 0.08 | 0.08 | **0.53** | 0.04 | 0.00 | 6.69 |
| **8** | Monitor the water quality | 0.00 | 0.04 | 0.02 | 0.04 | 0.04 | 0.08 | **0.57** | 0.06 | 0.02 | 0.11 | 6.83 |
| **9** | Improve governance mechanisms | 0.06 | 0.02 | 0.02 | 0.08 | 0.04 | 0.06 | 0.08 | 0.06 | **0.53** | 0.02 | 7.17 |
| **10** | Focus on Environmental Consciousness | 0.06 | 0.04 | 0.02 | 0.00 | 0.06 | 0.04 | 0.06 | 0.08 | 0.08 | **0.51** | 7.91 |

In general, there was a convergence of grades regarding the hierarchy among the 47 specialists who answered the survey, from a total of 124 members of the SFRBC. For 8 strategies out of the 10 selected, there was a concentration of more than half of the respondents (i.e., repetition >50), on a specific scale from 1 to 10 (bold values in Table 1). The exceptions were Strategy 3—*Increase the efficiency of production processes*—(44.4%) and Strategy 6—*Construction of Water Infrastructure*—(48.9%), which showed less dispersion among the dataset. These strategies can be considered more difficult to be agreed upon by the expert group consulted. Strategy 3—*Increase the efficiency of production processes*—, with a lower degree of consensus (44%), aims to improve the management of demand for water use, above all, to a greater efficiency than the methods used in irrigation (a sector that represents 70% of SFRB water consumption) [28] in sanitation, industry, and mining. Strategy 2—*Implement Water Pact*—concentrated the highest frequency of response in one range (62%); that is, 28 participants ranked it in grade 2, i.e., in second place of importance. This strategy was foreseen in the first SFRB planning instrument [32] and reaffirmed in the current SFRB Water Resources Plan 2016–2025 [28]. The planning instrument assumes allocation of water by sub-basin in each of the seven states of Brazil (Minas Gerais, Goiás,

Bahia, Pernambuco, Alagoas, Sergipe, and Distrito Federal) that make up the SFRB. Strategy 1—*River Basin Revitalization*—scored first place, with an average of 3.7; only 7 respoTdents out of a total of 47 (14%) rated it below or equal to 5 for this factor, demonstrating the importance of this strategy on a hierarchical scale as a top priority. The SRFBC claims there have been actions by the federal government to revitalize the SFRB since its creation in 2001. As Mello [33] observed, on the one hand, the S. Francisco River transposition project generated conflicts and controversies within the SFRBC; on the other hand, it was the driving and integrating element around a single theme, nationally designing the Committee and uniting the basin stakeholders around the theme of revitalization.

After compiling and analyzing the responses, the six strategies with the highest scores in the first round were selected, and then, in a second round, the SFRBC experts were consulted, by questionnaires, about their agreement with the responses. The results are shown in Table 2.

**Table 2.** Agreement degree bands displayed by the strategies in the first round of the Delphi method (*n* = 26).

| Strategies for Improving Water Security | 1st Round Grade | Agreement Degree | |
|---|---|---|---|
| | | Yes | No |
| 1st Implement a comprehensive River Basin Revitalization Program | 3.70 | 83.5% | 16.5% |
| 2nd Implement the Water Pact | 3.72 | 91.6% | 8.4% |
| 3rd Increase the efficiency of production processes | 3.91 | 91.6% | 8.4% |
| 4th Invest significantly in sanitation improvement | 4.59 | 83.5% | 16.5% |
| 5th Increase resilience in the face of hydrological disasters | 4.91 | 96% | 4% |
| 6th Construction of Water Infrastructure | 5.49 | 92% | 8% |

A general consensus was reached, with a degree of agreement greater than 80% among the 26 respondents who participated in the second round, converging to validate the results of the first phase. Strategy 1—*River Basin Revitalization*—, advocated by the panel of experts in this research as the main strategy for improving water security in the first round of the Delphi method, obtained 83.5% agreement, in line with the research hypothesis. We assumed that if river basin revitalization had emerged in first place as a guarantor of water security in the SFRB and with a high agreement rate (above 80% in the second round), the hypothesis should be considered validated. This result may be due to the multiple functions of revitalization, favoring natural water supply conditions and the integrity of ecosystems, with a positive impact on the population's life situation, on productive activities, and on disaster management, and also on minimizing potential conflicts.

In case of disagreement, the respondent could change their score, assigning another degree of hierarchy and justifying the change. It can be said that 1 of the 4 (16.5%) participants who disagreed with the score of Strategy 1—*River Basin Revitalization*—justified their decision by arguing: "(river basin) *revitalization is important, but the river and the basin should be preserved before*". Another respondent added as a strategy "*to implement a soil and water management program, through rural technical assistance*". The answers of the respondents that disagreed with the score of Strategy 1—*River Basin Revitalization*—showed their lack of knowledge about the term *river basin revitalization*, which already includes actions for environmental preservation and soil and water conservation to guarantee sustainability in the river basin [34].

The revitalization of the SFRB, proposed by the federal government of Brazil, presents a diversified set, which includes: preservation actions and environmental recovery; guaranteeing decent access to water; boosting economies with sustainable bases; prevention in risk areas; and basic sanitation [35]. Therefore, the concept of river basin revitalization is directly related to water security, and it is also related to the river restoration concept. River restoration is a relatively recent phenomenon, beginning in the 1970s and 1980s, particularly in response to water quality contaminations associated with industrial devel-

opment, as can be seen in the USA Clean Water Act of 1972 and in the European Union Water Framework Directive from 2000. Remarkable ground experiences at the river basin scale have been implemented on the River Rhine, the River Danube, and the River Thames since the 1980s and 1990s in Europe, and on the Cheonggyecheon River in South Korea in the 2000s [36]. In Switzerland, the Water Protection Act of 1991, revised in 2011, required the revitalization of rivers throughout the implementation of measures. Those measures were proposed for the restoration of natural functions of dammed or channeled ecosystems through infrastructures being built that were directly linked with flood protection [37]. In general terms, river restoration includes any action aimed at improving the health of a river and, consequently, ecosystem services [38]. Strategy 5—*Increase the resilience of the basin*—presented the greatest consensus. This result emerged in the face of hydrological disasters, directly connected to the adaptation processes of critical climatic events, which is why it was viewed as a priority by the SFRBC [28].

The concordance on the choice of strategies may reflect the "water crisis", which has affected the water supply as a result of a lack of rain. This phenomenon has reached several regions of Brazil, especially from 2012 to 2017, with special severity in the semi-arid region through which the SFRB passes. In fact, the SFRB has recorded precipitation values below the historical average, with a significant reduction in water volumes flowing to the reservoirs of hydroelectric dams in the basin. As a main consequence, the lowest levels of water storage have been registered, putting the possibility of continuing to maintain multiple water uses at risk [22]. It is not only droughts that have had an effect; floods, pollution, and dam failure have had devastating consequences for human development around the SFRB. These extreme water insecurity phenomena embrace a more inclusive approach to the water security concept: protection and disaster risk prevention. According to Grey and Sadoff [7], who highlighted this dual character of water, the concept of water security should consider the availability of an acceptable quantity and quality of water for health, livelihoods, ecosystems, and production, safeguarding the security of people, environments, and regional economies. Strategy 6—*Construction of Water Infrastructure*—was ranked sixth. This corresponds to the main strategy of the federal government integrated in the National Water Security Plan [22]. This Plan includes a set of structural interventions that guarantee lasting results, namely: construction of dams for water regulation for human supply or multiple uses; construction of infrastructures for flood control and for water conduction and derivation.

The GWP [39] mentions that infrastructures (such as dams) are necessary to meet the current and projected demands for the future and must be part of the strategy to achieve water security of a given hydrological unit. However, infrastructures must be carefully planned and designed using technologies that minimize environmental impacts. If, on the one hand, dams reduce the variability of water availability in a territory, on the other hand, they also cause significant environmental impacts, such as the interruption of natural water flows on which aquatics ecosystems and their own biodiversity depend. The regional impacts of the creation of a large surface of water should also be considered. Within this concept, it is important to mention nature-based solutions (NBS) to minimize the contemporary problems of water supply according to quality standards, while guaranteeing the ecosystem's ecological integrity [40]. In this context, restoration of aquatic ecosystems and river basin revitalization become even more significant. Regarding complementary responses, other experts cited as a main strategy the "*systemic view and permanent dialogue among the various actors*", plus "*implementing a consistent and effective information system*".

The results of the consultation process with the SRFBC were in accordance with the proposal of Grey and Sadoff [8], who defended the logic of the three Is to achieve water security: Investment, Institutions, and Information. These authors analyzed the role of the economic history of water in the development of nations and at the national level, demonstrating that the path to water security should combine high investment with robust institutions and adequate infrastructure. That is, depending on the punctuated strategies, investments in revitalization actions and in infrastructure to access, store, and reuse water,

as well as in robust institutions and the information and capacity to predict, plan, and deal with climate variability, would also be necessary.

The six most voted strategies in this research were aligned with the three pacts defined for the Water Resources Plan to prosper [28]: (1) the Waters Pact, which takes the negotiated allocation of water between the states that are part of the SFRB; (2) the Legality Pact, in which the instruments for water resources management in the basin are universalized [41]; and (3) the River Basin Revitalization Pact, so that actions to effectively revitalize the basin become a reality in terms of resources and political will. Based on political participation, river basin committees are already considered innovative because, in addition to being consultative, they collectively deliberate on water management in a shared way with the government, which drives an approach that favors social participation.

Another relevant piece of information about the current status of the concept and its relation to the importance of revitalizing the SFRB is the UN's approval in 2019 [42] of the new International Decade on Ecosystem Restoration (2021–2030). The objective of this decade is to prevent, interrupt, and reverse the degradation of ecosystems around the world as a proven measure to combat the climate crisis and improve food security, water supply, and biodiversity. The conservation of biodiversity and the sustainable use of natural resources are essential to achieve this goal. This decade, a global call to action will bring together political, financial, and scientific support to expand restoration initiatives, and the case of the SFRB could be one of them.

### 3.2. Symbiotic Process: Water Security Versus River Basin Revitalization

Symbiosis, from the Greek words *syn* (together) and *bios* (life), is a term originally utilized by biological sciences to describe an ecological relationship between those who are part of a very close association but belonging to different species. In this ecological relationship, organisms need one another to survive. Likewise, river basin revitalization, with the maintenance of natural water sources, is directly related to the water security concept. Thus, the interdependence between river basin revitalization and water security is closely linked in a symbiotic process. This metaphor around the word symbiosis has also been used to state the relationship between water security and integrated water resources management [3]. We use it here because it fits the analogy well. This reasoning can be applied in a similar way to the SFRB because the premise of improving water security has always been a key inclusion factor for efforts towards the São Francisco River basin revitalization, based on two political aspects: (1) increasing the supply of water for the basin, referred to in the original design of the SFRP [43]; (2) the recurring association with the São Francisco River Transposition Project. It is important to highlight the fact that this project (transposition) is full of contradictions. It is supposed to induce water security in drier areas of the Brazilian northeast region with the construction of canals that would supply reservoirs and intermittent rivers, despite the possibility of negatively affecting ecological and social relations in the SFRB itself.

Beyond identifying symbolic processes between both concepts—river basin revitalization [23] and water security [44]—a concept map was elaborated with the Delphi method results, based on the assumptions presented in Table 3.

**Table 3.** Comparison elements between river basin revitalization and water security.

| Categories | River Basins Revitalization | Water Security |
|---|---|---|
| Propagation | Debate over the São Francisco River (SFR) Transposition Project | 2nd World Water Forum |
| Starting Focus | Water offered | Water quantity and quality |
| Key Concept | Recovering, conserving, and preserving environmental processes | Capacity to safeguard sustainable access to water in adequate quantity and quality |

**Table 3.** *Cont.*

| Categories | River Basins Revitalization | Water Security |
|---|---|---|
| Finality | (1) Sustainable use of natural resources; (2) Socioenvironmental conditions improvement; (3) Water quantity increase and quality improvement for multiple uses. | (1) Guarantee livelihood, well-being, and economic development; (2) Protection against pollution and water-related disasters; (3) Ecosystems' preservation in a peaceful and stable political environment. |
| Components | (1) Planning and information; (2) Environmental education; (3) Soil use and protection; (4) Environmental sanitation; (5) Sustainable economics. | (1) Human supply; (2) Economic productivity; (3) Ecosystem preservations; (4) Risk management; (5) Political stability. |
| Final goal | Improve water-related conditions for human well-being | |

The concept map (Figure 2) was chosen because it facilitates the visualization of themes and their interconnections, as well as the achievement of results, from the following question: What is river basin revitalization's place in relation to water security?

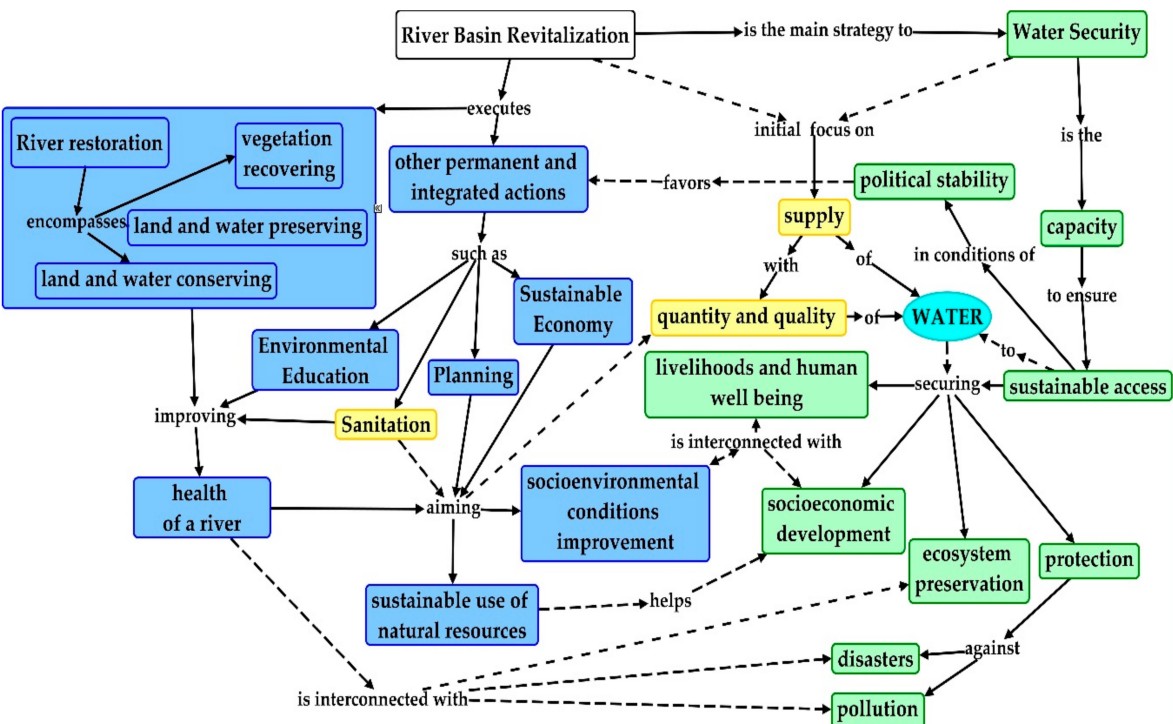

**Figure 2.** Concept map presenting the interdependence between river basin revitalization (blue boxes) and water security (green boxes). The relationships between them are represented by lines that link the boxes and by related words (connectors) that describe the nature of the relationship that binds them. The yellow boxes show the link to safe water components (supply, quality and quantity, sanitation), which corresponds to the ancient safe water source concept, focused on sanitation and hygiene.

In an integrative approach, the concept map structured in Figure 2 places the concept of river basin revitalization in relation to water security, therefore demonstrating a clear convergence between these themes. Additionally, the concept map shows how the revitalization concept is broader than the restoration concept. The map depicts river basin revitalization as a process linked to environmental recovery (of natural vegetation), conservation, and preservation of ecosystems through permanent and integrated actions,

such as: (1) Planning; (2) Environmental education; (3) Soil protection and use; (4) Sanitation; (5) Sustainable Economy. All these actions, acting together, acquire an amplifying effect that improves water quantity and quality, and consequently, socioenvironmental conditions and the sustainable use of natural resources. Water security, on the other hand, is portrayed as the ability to ensure sustainable access to water (in terms of quantity and quality) to guarantee: survival and human well-being; socioeconomic development; protection against pollution and disasters; the preservation of ecosystems in conditions of peace and political stability. The dashed lines establish a cross-relation, illustrating how the two themes (revitalization and water security) intersect in a more direct way. Figure 2 presents the main cross-relationships: socioenvironmental conditions' improvement connected to socioeconomic development and livelihoods and human well-being; river health connected to ecosystem preservation and also to protection against pollution and disasters. The fifth component—political stability—of the water security concept emerges and is connected to the capability to develop integrated actions and plans in the basin.

On the map, the arrows represent the direction of the conceptual relationship that drove the formulation of the hypothesis to be tested by the Delphi method, including SRBC members' opinions about the main strategies for achieving water security. As mentioned before, as a consensus, river basin revitalization was identified as the most important factor related to water security. The concept map summarizes the main details presented in Table 3, emphasizing the categories "Finality" and "Components" as the key elements that define the symbiotic process between the two concepts. Therefore, considering that natural resources need to be conserved, preserved, and recovered, we postulate that river basins should be revitalized to meet the multiple uses of water on a sustainable basis, which means the achievement of water security.

In the absence of an agreed criteria regarding the implementation and measurement of the results of revitalization [45], we defend the revitalization concept in line with the objectives of water security. The incorporation of the water security concept in the SFRP has enormous potential to produce solutions that benefit multiple sectors and enhance the effectiveness of implemented actions. Water security can provide a framework of goals for river basin revitalization, which is understood as a complex process. Water security, despite not being at the core of the mission of river basin revitalization actions, is one of its parts. On the other hand, river basin revitalization comes close to the broader concept of environmental security, where water security is one of its main results, focused on practical actions.

We also recognize that the construction of water infrastructures can be essential for achieving the water security objective (related to the guarantee of water for human beings). However, it is necessary to go beyond this, towards the concept of river basin revitalization and ecosystem preservation.

## 4. Conclusions

Most of the opinions of the SFRBC members argue that priority should be given to revitalization measures, which, in turn, are reflected in the state of water security. This result corroborates the research hypothesis. Although the hypothesis proven in this research may seem "common sense", what is seen from the actions put into practice by the government sector in the SFRB is a postponement of revitalization actions as a main strategy. As we have pointed out, in Brazil, the traditional idea prevails that water security is linked exclusively to large water infrastructure projects to meet increasing demands for water use. An example of this traditional approach is the Transposition Project of the São Francisco River being an integrated part of the National Water Security Plan [22]. Unfortunately, this water transfer project does not consider actions for basin revitalizing. Indeed, like the controversial Transposition Project, most of the actions that make up the National Water Security Plan do not consider biodiversity conservation and ecosystem preservation.

Our results indicate that river basin revitalization should be considered within the scope of the actions, strategies, and investments of the Water Security Plan. For this reason,

it is urgent to reconcile the use of water, ecological integrity, and ecosystem services through the comprehensive concept of water security. The exercise to approach these two agendas, assuming they are symbiotic, arrives at the right time. In institutional and political terms, since 2019, these two agendas (use of water and ecological integrity) have been under the responsibility of the Ministry of Regional Development, after leaving the responsibility of the Ministry of the Environment. It is expected that actions and projects could be developed in an integrative way, respecting ecological processes and environmental quality criteria while regional development and social inclusion and human rights are promoted to advance the SFRB's sustainability.

Finally, future investigations could be done in a different direction, and try to identify the differences in the prioritization of the SFRB Revitalization Program's Strategies by stakeholders. This may help to analyze possible influences of water users on the Strategy's development and better visualize ongoing disputes in the public sphere.

### Institutional Review Board Statement

The study was conducted according to the guidelines of the Declaration of Helsinki and approved by the Research Ethics Committee at the University of Brasília - process number 98071318.0.0000.5540.

**Author Contributions:** Conceptualization, L.A.d.S.R., M.M.; C.H.S.; methodology, L.A.d.S.R., M.M.; C.H.S.; software, L.A.d.S.R. and C.H.S.; formal analysis, L.A.d.S.R. and C.H.S.; investigation, L.A.d.S.R. and C.H.S.; writing—original draft preparation, L.A.d.S.R.; M.M.; C.H.S.; writing—review and editing, L.A.d.S.R.; M.M.; C.H.S. All authors have read and agreed to the published version of the manuscript.

**Funding:** This work was supported by the research project INCT/Odisseia-Observatory of socio-environmental dynamics: sustainability and adaptation to climate, environmental and demographic changes under the National Institutes of Science and Technology Program (Call INCT – MCTI/CNPq/CAPES/FAPs n.16/2014) with financial support of CNPq, CAPES and FAPDF. The edition cost was co-funded by the European Union through the European Regional Development Fund, included in the COMPETE 2020 (Operational Program Competitiveness and Internationalization) through the ICT project (UIDB/04683/2021).

**Informed Consent Statement:** Informed consent was obtained from all subjects involved in the study.

**Data Availability Statement:** Not applicable.

**Acknowledgments:** The authors thank São Francisco River Basin Committee (SFRBC), as well as SFRBC members, who were consulted in two successive rounds of questionnaires, allowing the obtention of the analyzed and presented data in this study.

**Conflicts of Interest:** The authors declare no conflict of interest.

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
