# Peer review of "Water Security and River Basin Revitalization of the São Francisco River Basin: A Symbiotic Relationship"

_water, doi:10.3390/w13070907_

Round 1

Reviewer 1 Report

In the manuscript, the author analyzed the relationship and concept between Water Security and River Basin Revitalization in the São Fran-cisco River Basin,and got some Symbiotic Relationships results. I think the method and analyses are right and important for river management. I think it is accepted.

Some comments

  1. Correspondence: to whom correspondence should be addressed? Please list the Corresponding author’s email
  2. The index should be designed according to São Fran- cisco River Basin in table 1. or the research can be suitable for all rive basins in the world.
  3. Give the method flowchart.

Reviewer 2 Report

Thank you for the paper. Overall it is good paper and the authors attempt to conceptualize River Basin Revitalization based on the concept of water security. I have some minor concerns. For clear understanding of the methodology, I would suggest the authors to explain the methodology using a flowchart. Did the authors check how the prioritizations might have varied based on the characteristics of the respondents? These findings will be interesting to read. In addition, community participation is an important component of water security and also play an important role in river basin management. Where is community participation in the identified strategies presented in Table 1? I would suggest the authors modify the discussion and the conclusion to emphasize the policy implications of the study.

Reviewer 3 Report

Dear Authors,

I am writing this to submit my comments on your research article with the following details.

Manuscript title: Water Security and River Basin Revitalization of the São Francisco River Basin: A Symbiotic Relationship

Manuscript Number: water-1115250

Journal Submitted: Water

Specific Comments:

Title:

The title is OK.

Abstract:

Please use the past tense.

L 12-14: What was the sample size for the interview?

No conclusions were provided.

The keywords are too long and less in number.

Introduction:

L 68: Mention the name of the authors.

The abbreviations must go in the brackets.

There is a lack of connection and coherence among the paragraphs.

L 57-62: The concept of terrorism appears to be a bit far connected with this concept. Further, you may consider adding other threats linked with water security as well.

The objectives are not described adequately.

Materials and Methods:

Please provide more details on the Delphi method.

How many experts did you consult?

L 134-134: This is rather a repetition or does not belong here.

Please split this section into subsections and avoid jargon.

How did you finalize the questions, and what criteria did you use?

Did you explain the background of the strategies used?

What was the expertise level of the persons you took help during the questionnaire filling?

Results:

The results are weak in their description, and the authors need to restrict the report only and no need to explain them here.

Further, after seeing the quality of the result, I suggest combining the results with the discussion section.

Discussion:

The discussion is the most substantial part of this manuscript.

Figures and Tables:

Fine.

Conclusions

Too long. It should be reduced in content.

References:

Good.

Round 2

Reviewer 2 Report

Thank you for carrying out the corrections carefully. 

I found the paper highly improved and acceptable for publication.

Reviewer 3 Report

The authors have greatly improved the manuscript during revisions, therefore, I believe no more changes could be suggested. Kudos.